# Stable demographic ratios of haploid gametophyte to diploid sporophyte abundance in macroalgal populations

**Kazuhiro Bessho** ⬤ *

Saitama Medical University, Saitama, Japan

* besshokazuhiro.research@gmail.com

## Abstract

Macroalgal populations often consist of free-living haploid (gametophyte) and diploid (sporophyte) stages. Various ecological studies have been conducted to examine the demographic diversity of haploid-diploid populations with regard to the dominant stage. Here, I relaxed the assumption of classical research that the life history parameters of haploids and diploids are identical and developed a generalized haploid-diploid model that explicitly accounts for population density dependence and asexual reproduction. Analysis of this model yielded an exact solution for the abundance ratio of haploids to diploids in a population in which the ratio is determined by the balance of four demographic forces: sexual reproduction by haploids, sexual reproduction by diploids, asexual reproduction by haploids, and asexual reproduction by diploids. Furthermore, the persistence of a haploid-diploid population and its total biomass are shown to be determined by the basic reproductive number ($R_0$), which is shown to be a function of these four demographic forces. When $R_0$ is greater than one, the haploid-diploid population stably persists, and the ploidy ratio obtained by the analytical solution is realized.

## Introduction

Sexual reproduction in eukaryotes leads to the alternation of two nuclear phases, namely the haploid ($n$) and diploid ($2n$). In many plants, algae, and fungi both ploidy stages develop into adult free-living organisms and as such, a biphasic life cycle is observed [1–7]. Many macroalgae alternate between a free-living haploid generation (the gametophytes) and the free-living diploid generation (the sporophytes).

The life cycle diversity has been interesting in the context of how the evolution from haploidy to diploidy originally occurred [8, 9]. For example, the hypotheses that the diploids have an advantage in masking deleterious mutations [9–11], that the diploids evolve faster [12, 13], and that the haploids have an advantage in poor-nutrient conditions [14–16], have been considered.

Population geneticists have focused primarily on life cycle evolution, whereas many phycologists have been interested in the descriptive study of the macroalgal life cycles. Classically, the

**Data Availability Statement:** All relevant data are within the paper and its Supporting Information files.

**Funding:** This research was supported by Grants-in-Aid from the Japan Society for the Promotion of

Science to KB (19K16225; 22K06407).The funders had no role in study design, data collection and analysis, decision to publish, or preparation of the manuscript.

**Competing interests:** The authors have declared that no competing interests exist.

type of alternation of generations between two distinct free-living stages is classified as heteromorphic (morphologically distinct from one another) or isomorphic (seemingly identical to each other) [17]. Species with the heteromorphic life cycle show large phenotypic differences (e.g., macroscopic versus microscopic, erect versus crustose). These species also often show different seasonal appearances and habitats [18]. In contrast, the two stages of species with isomorphic life cycle tend to be observed simultaneously and sympatrically [19–23].

Because these studies were based on laboratory observations, there was interest in field research. Especially in species with the isomorphic life cycle, gametophytes and sporophytes (tetrasporophytes in red algae) are often observed simultaneously, and their abundances were investigated. Field studies of the ratio of haploid to diploid abundance have shown both haploid-dominant [20, 21, 23] and diploid-dominant [19, 22] populations exist. Different patterns of haploid or diploid dominance have been observed for different populations of the same species, and even over the same population depending on the special location or season [24–27]. In their demographic researches, the frequency of each stage [23, 28], the haploid-to-diploid (H:D) ratio [29], and the gametophyte-to-sporpohyte (G:T) ratio [21, 30, 31] were discussed.

Theoretical models have been developed to explain the variety of haploid versus diploid dominance patterns [28, 30–38]. Many studies are based on numerical calculations, and few studies derive simple analytical solutions. In particular, the classical study of Thornber and Gaines [28] presented an analytical solution only for the case where haploids (gametophytes) and diploids (tetorasporophytes) have identical survival and growth rates (i.e., ploidy abundance ratio = $\sqrt{2} : 1$).

Although in their classical study, Littler et al. [39] reported the functional similarity in physiological or ecological performances between gametophyte and sporophyte in an isomorphic species, many recent empirical studies have reported various types of functional differences between haploids and diploids. Specifically, chemical composition [40], resistance to physical stress [41], fecundity [22, 28], survivorship [29, 42, 43], resistance to predators [44], resistance to epiphyte infection [45], growth rate [46–48], dispersal ability of the reproductive cells [49], and viability of the reproductive cells [50] are reported. Based on these facts, this article analyses a mathematical model that explicitly considers the differences in life history parameters of haploid gametophytes and diploid sporophytes, and aims to express stable ploidy ratios in a haploid-diploid population using a mathematical expression as simple as possible.

In recent years, in addition, the importance of asexual reproduction, which deviates from the normal reproductive process in which gametophytes reproduce sporophytes and sporophytes reproduce gametophytes (see next section), has been pointed out [51–53]. Unfortunately, theoretical studies that consider asexual reproduction are limited. Among the few studies, [53] analyze the demographic equilibrium state of a population genetic model that considers sporogenesis, and Vieira and Santos [31, 35] analyze the population structure yield by a matrix model that considers clonal reproduction (i.e., vegetative growth of new fronds from the same holdfast). In addition, this study explicitly considers density dependence, which many previous studies have omitted for simplicity. Many previous studies (e.g., [31]) define the ploidy ratio by the right eigenvector in the matrix model, where the density dependence is ignored and the population size diverges to infinity. In reality, the population size should remain finite and the stable ploidy ratio of this population is important. Vieira et al. [38] develop the individual-based models to investigate population dynamics that consider the realistic density dependence, but the results are complex and analysis of a simpler model is required. Analyzing the model, I investigate the ploidy ratio in a haploid-diploid population with asexuality and if the ploidy ratio is in agreement with the results of previous studies when the parameters are constrained accordingly.

## Life cycle of macroalgae

In this section, I briefly review the life cycle diversity in macroalgae (see also [3, 6, 7, 17, 54]), which is the prerequisite knowledge of the models. A diagram of the reproduction process assumed in the model is shown in Fig 1A.

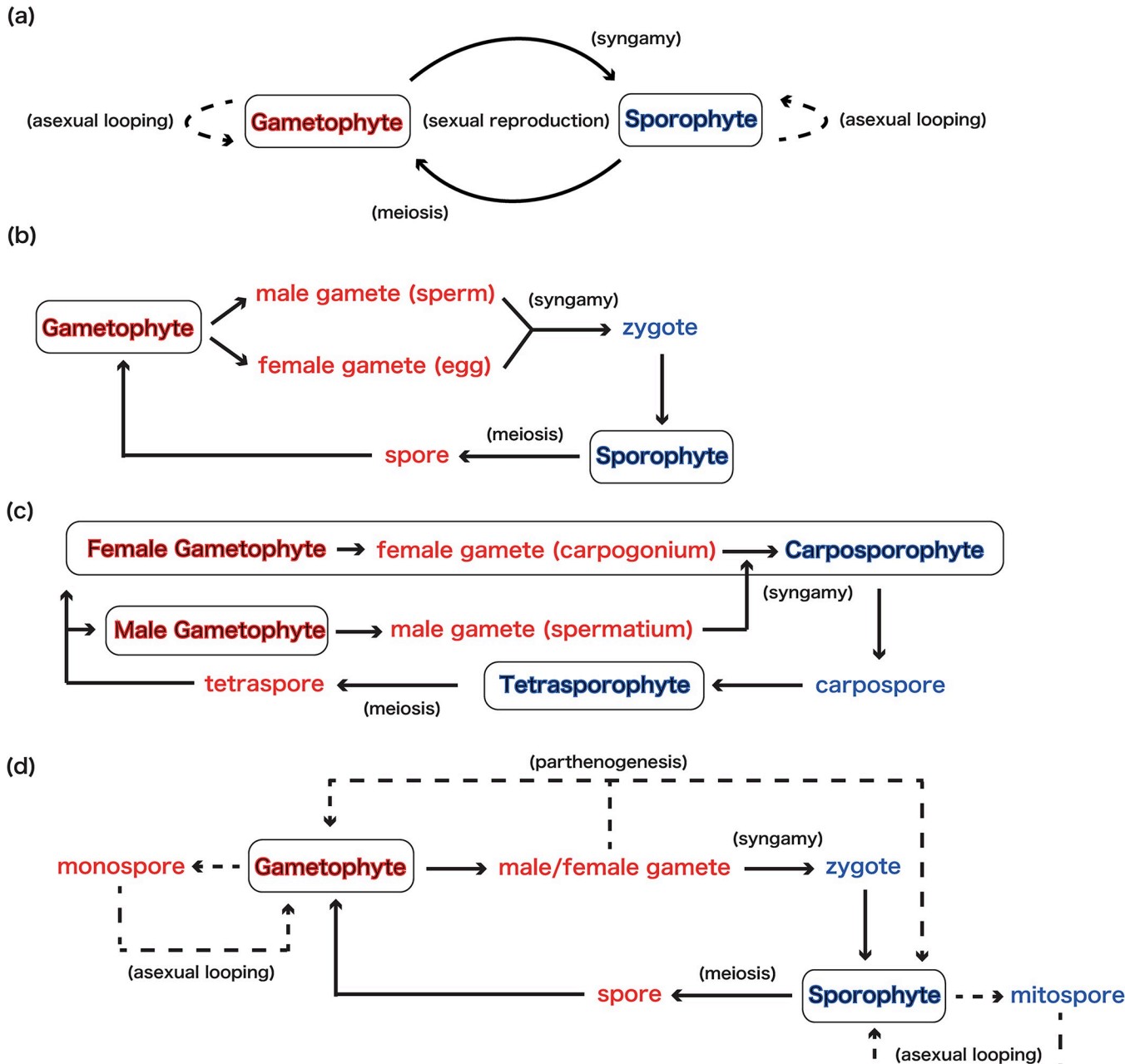

**Fig 1. Diagrams of macroalgal life cycles.** (a) A diagram of the reproduction process assumed in the model. (b) A diagram of the alternation of generations between haploid gametophyte and diploid sporophyte (haploid-diploid life cycle). In this diagram, monoecious is assumed. (c) A diagram of the general life cycle of Florideophyceae (triphasic life cycle). (d) A diagram of the asexual reproduction in macroalgae. Red indicates the haploid phase and blue indicates the diploid phase. The circled states are free-living stages.

## Alternation of nuclear phases and generations

In sexual eukaryotes, states characterized by one set of chromosomes (1n, haploid) alternate with states characterized by two sets of chromosomes (2n, diploid) through meiosis and zygosis. This "alternation of nuclear phases" often results in the alternation of individuals that exhibit different reproductive patterns during their life cycle, which is referred to as the "alternation of generations". In many macroalgae, haploid multicellular individuals and diploid multicellular individuals appear as part of this alternation of generations (haploid-diploid life cycle). Generally, the former is called "gametophytes" because they reproduce by gametes, and the latter is called "sporophytes" because they reproduce by spores (Fig 1B).

I note that, however, the alternation of generations is not an essential phenomenon in macroalgae. For example, individuals belonging to Fucales, a group of brown algae, are diploid. When they mature, gametogenesis and syngamy occur continuously, and the life cycle is completed as the diploid zygote develops into a multicellular organism again (diploid life cycle). In other words, they do not exhibit alternation of generations.

## Heteromorphic and isomorphic life cycles

When we focus on the appearance of macroalgal gametophytes and sporophytes, they are often very different in morphology. Extreme examples include species of Laminariales, in which macroscopic sporophytes alternate with microscopic gametophytes, and the species of Scytosiphonales, in which erect gametophytes alternate with crustose sporophytes. This mode of alternation of generations is called "heteromorphic" alternation of generations. On the other hand, (especially immature) gametophytes and sporophytes are often so similar that they are indistinguishable (e.g., Ulva), and such an alternation of generations is called "isomorphic" alternation of generations [17]. The life cycle characterized by the heteromorphic (isomorphic) alternation of generations is called the heteromorphic (isomorphic) life cycle.

I note that the difference between gametophytes and sporophytes is sometimes less extreme. In the brown algae Ectocarpus, for example, both gametophytes and sporophytes are small and filamentous, but there is a substantial morphological difference between them. This species is classified as "heteromorphic (near-isomorphic)" [51]. The accumulation of empirical studies showing that gametophyte and sporophyte (or tetrasporphyte of red algae) phenotypes differ (morphologically, ecophysiologically, or biomechanically) calls into question the classical classification of the life cycle into heteromorphic and isomorphic categories.

## Triphasic life cycle in red macroalgae

When we consider macroalgal life cycle, we generally assume alternation of generations between haploid gametophytes and diploid sporophytes (biphasic life cycle). Among red algae, however, many species from the Florideophyceae are known to exhibit more complex life cycles. Here, syngamy occurs when a haploid sperm without flagella (spermatium) attaches to haploid female gametes (carpogonium) that attaches on the female gametophyte. This fertilized cell develops into diploid multicellular structure, the carposporophyte. First, the carposporophyte produces diploid carpospores by somatic cell division. Only later they are released and these develop into free-living diploid sporophytes, tetrasporophytes. Finally, the tetrasporophyte produces haploid tetraspores by meiosis, and these develop into gametophytes to complete their life cycle. Their life cycle involves three "phases" of haploid gametophyte, diploid carposporophyte, and diploid tetrasporophyte, so this can be called a "triphasic" life cycle [55].

Because the ploidy of carposporphyte (diploid) is different from gametophyte (haploid), the caposporophyte is recognized as a distinguishable phase, but is nutritionally dependent (parasitic) on the female gametophyte [56]. Hence, if we consider that carposporophytes have little effect on

the ecological properties of the species, the triphasic life cycle of red algae can be approximated as a biphasic life cycle alternating only between gametophyte and tetrasporophyte (Fig 1C).

### Asexual reproduction in macroalgae

In classical ecology and evolutionary researches about the macroalgal life cycle, researchers assume obligate sexuality, with gametophytes reproducing sporophytes and sporophytes reproducing gametophytes (e.g., [9, 57]). However, many macroalgae exhibit complex reproductive strategies, including asexual reproduction (e.g., [51]). Examples include parthenogenesis, in which gametes that fail to syngamy develop into gametophytes, asexual multiplication of gametophytes by monospores, asexual multiplication of sporophytes by mitospores, and vegetative growth (e.g., the growth of new fronds from the same holdfast or from broken fragments) (Fig 1D).

For simplicity, this article models asexual reproduction, as processes in which haploid gametophytes reproduce haploid gametophytes and diploid sporophytes reproduce diploid sporophytes by special reproductive cells (e.g., mitospores). I note, however, that it is also known that "haploid sporophytes" reproduced by haploid gametophytes and "diploid gametophytes" reproduced by diploid sporophytes are observed in empirical conditions [58, 59].

## Model

### Population dynamics without density dependence

First, the population dynamics of a haploid-diploid population without density dependence is analyzed by updating the matrix model [28]. The haploid ($H$) and diploid ($D$) densities are forecasted to time $t+1$ from their realized densities at time $t$ and their vital rates:

$$\begin{pmatrix} H(t+1) \\ D(t+1) \end{pmatrix} = \begin{pmatrix} 1 + \gamma_H b_H a_H - m_H & \gamma_H b_D(1 - a_D) \\ \gamma_D \dfrac{f b_H}{2}(1 - a_H) & 1 + \gamma_D b_D a_D - m_D \end{pmatrix} \begin{pmatrix} H(t) \\ D(t) \end{pmatrix}. \tag{1}$$

Here, the survivorship of haploid and diploid reproductive cells (tetraspores and carpospores for red algal species) while suspended in the water column is $\gamma_H$ and $\gamma_D$, respectively; the fecundity of the haploid and diploid stages per individual is $b_H$ and $b_D$, respectively; and the mortality rates of haploids and diploids are $m_H$ and $m_D$ (adult survival is $1-m$), respectively. The $b_H$ is the fecundity rate of haploid females (production of carposporophytes for red algal species). However, not all haploids are females as some are males. Assuming a 1:1 sex ratio, 1/2 of the haploids are females (or we can also assume the species is monoecious). Hence, in the matrix model, haploid fecundity is $b_H/2$. Then, comes the reproductive cost (the probability of fertilization success) $f$, leading to $f b_H/2$. In the following analysis, I assume this cost (cost of sex) is defined as $\sigma = f/2$. Fractions $a_H$ and $a_D$ of the reproductive outputs of the haploid and diploid stages are asexual, skipping the sexual loop and developing directly into haploids and diploids (sporogenesis). When $a_H = a_D = 0$, Eq (1) describes the case of a fully sexual (obligate sexual) system [28]. The dominance patterns of two stages following spatial or seasonal patterns are often observed in the field, however for tractability, constant life history parameters and a well-mixed population are assumed.

### Population dynamics with density dependence

A density-dependent population of macroalgae that consists of haploids and diploids can be described by the following differential equations:

$$\frac{dH}{dt} = \gamma_H \phi(t) - m_H[1 + \delta(H(t) + D(t))]H(t), \tag{2A}$$

$$\frac{dD}{dt} = \gamma_D \psi(t) - m_D[1 + \delta(H(t) + D(t))]D(t), \tag{2B}$$

$$\phi(t) = b_H a_H H(t) + b_D(1 - a_D)D(t), \tag{2C}$$

$$\psi(t) = \sigma b_H(1 - a_H)H(t) + b_D a_D D(t), \tag{2D}$$

where $\phi(\psi)$ indicate the number of haploid (diploid) offspring at time $t$. Here, the mortality rate is assumed to increase with the biomass of the population ($H(t)+D(t)$) at rate $\delta$. When population dynamics are modeled with density dependence, other modes of action, besides increasing mortality, should be considered (e.g., decreasing fecundity). Other types of density dependence are considered in Appendix D in S1 Appendix.

## Results

### Ploidy ratios with a density-independent model

With a density-independent model, the population size increases exponentially. In this model, the right leading eigenvector, which gives the stable distribution of haploid and diploid abundances in the population, is conceptually equivalent to the stable age distribution in the Leslie matrix model.

The eigenvalues and right eigenvectors for the full model are provided in the S1 File. The haploid frequency in the population, $\rho_H$, can be calculated as follows:

$$\rho_H = \frac{(m_D - m_H) + w_H^A m_H + w_D^A m_D - 2(w_D^S m_H + w_H^A m_D) + \sqrt{4 w_H^S w_D^S m_H m_D + [(m_D - m_H) + w_H^A m_H - w_D^A m_D]^2}}{2[(m_D - m_H) + w_H^A m_H + w_H^S m_D - w_D^A m_D - w_D^S m_H]}, \tag{3}$$

where the fitness components of haploids and diploids are defined as,
$w_H^A = (b_H a_H \gamma_H)/m_H$, $w_D^A = (b_D a_D \gamma_D)/m_D$, $w_H^S = [b_H(1 - a_H)\sigma \gamma_D]/m_D$, and $w_D^S = [b_D(1 - a_D)\gamma_H]/m_H$, and the superscripts $S$ and $A$ refer to sexual and asexual reproduction, respectively. Note that the fitness components for sexual reproduction ($w_H^S$ and $w_D^S$) is defined by the mortality of different ploidy individuals. This is because the reproductive cells for sexual reproduction reproduced by individuals of focal generation (e.g., haploid) will develop to the opposite generation (e.g., diploid), so mortality should be different from the focal individual.

If the species has an "ideal" isomorphic life cycle and exhibits full sexuality (obligate sexuality), the number of parameters can be reduced as follows: $\gamma_H = \gamma_D$, $m_H = m_D$, $b_H = b_D$, $a_H = 0$, and $a_D = 0$. In this case, the haploid ($\bar{H}$) to diploid ($\bar{D}$) ratio in asymptotic density is,

$$\bar{H} : \bar{D} = 1 : \sqrt{\sigma}. \tag{4}$$

If the cost of fertilization is ignored ($f = 1$), then $\sigma = 1/2$ and Eq (4) is the same as the classical ploidy ratio of $\sqrt{2} : 1$ ([28], p. 1664).

If the fecundity of haploids is assumed to be twice that of diploids, because the haploid cell size tends to be half the diploid cell size, then $\gamma = \gamma_H = \gamma_D$, $m = m_H = m_D$, and $b_H = 2b_D$, and the ploidy ratio is,

$$\bar{H} : \bar{D} = 1 : \sqrt{2\sigma}. \tag{5}$$

If the cost of fertilization is ignored ($f = 1$), then $\sigma = 1/2$ and the ploidy ratio in this case is 1:1.

## Ploidy ratio of a fully sexual species with density dependence

In the density-independent model, population size increases infinitely, which is unrealistic. To avoid this problem, the density-dependent model is analyzed next. For tractability, the model for a species that exhibits full sexuality ($a_H = a_D = 0$) is analyzed first. In this case, strict solutions for both biomass ($T = \hat{H} + \hat{D}$) and haploid frequency ($\rho_H = \hat{H}/(\hat{H} + \hat{D})$) are obtained at population equilibrium ($\hat{H}, \hat{D}$),

$$T = \frac{\sqrt{w_H w_D} - 1}{\delta} \text{ and } \rho_H = \frac{\sqrt{w_D}}{\sqrt{w_H} + \sqrt{w_D}}. \tag{6}$$

Here, the fitness of haploids and diploids is defined as $w_H = (\sigma b_H \gamma_D)/m_D$ and $w_D = (b_D \gamma_H)/m_H$, respectively. Note that the mortality rate in the fitness of haploids ($w_H$) should be defined by the diploid mortality ($m_D$), and vice versa. This is for exactly the same reason as the definition of the fitness components in the full model ($w_H^s$ and $w_D^s$).

This haploid frequency ($\rho_H$ in Eq 6) is equivalent to the haploid-diploid equilibrium frequency in the Wright-Fisher model in Bessho and Otto ([60]; their Eq (D2)). Furthermore, the haploid frequency, $\rho_H$, can also be represented as follows:

$$\rho_H = \frac{\sqrt{(b_D \gamma_H)/m_H}}{\sqrt{(\sigma b_H \gamma_D)/m_D} + \sqrt{(b_D \gamma_H)/m_H}} = \frac{\sqrt{[(b_D \gamma_H) m_D]}}{\sqrt{[f(b_H \gamma_D) m_H]/2} + \sqrt{[(b_D \gamma_H) m_D]}}.$$

This haploid frequency is the same as that in the Moran model in Bessho and Otto ([60]; their Eq (A8b)).

Note that when the mortalities of haploids and diploids are equal, $\rho_H/\rho_D = \sqrt{w_D}/\sqrt{w_H}$ becomes $\sqrt{b_D \gamma_H}/\sqrt{\sigma b_H \gamma_D}$, which is the gametophyte-to-sporophyte ratio in Fierst et al. ([30], their Eq (15))

## Parameter dependence in species with obligate sexuality

The basic reproductive number of species with obligate sexuality is the geometric mean of haploid fitness and diploid fitness, $R_0 = \sqrt{w_H w_D}$ (Appendix C2 in S1 Appendix). Thus, the equilibrium biomass calculated using this measure is,

$$T = \frac{R_0 - 1}{\delta} \tag{7}$$

Thus, the biomass (total population size) holds constant at a positive value at equilibrium when $R_0 > 1$. Furthermore, when $R_0 > 1$, the trivial equilibrium, $(0, 0)$, is unstable and the haploid-diploid population exists stably as in Eq (6) (Appendixes B3 and B4 in S1 Appendix).

Fig 2 illustrates the parameter dependence of the haploid frequency $\rho_H$ and equilibrium biomass, scaling the population size by $\delta$ in Eq (7) (i.e., $\tilde{T} = T\delta = R_0 - 1$; hereafter, the biomass measure). The ploidy ratio and biomass both depend on haploid fitness ($w_H$) and diploid fitness ($w_D$). Because haploids reproduce diploid individuals and vice versa, haploids are dominant in a population with higher diploid fitness. Note, however, that fitnesses are defined by the mortality of opposite ploidy individuals (see the definition). For example, a high haploid mortality rate (high $m_H$) reduces the diploid fitness (low $w_D$) and consequently increases the frequency of diploids (low $\rho_H$).

## Ploidy ratio in asexually reproducing species under density dependence

For asexually reproducing species, strict solutions for biomass ($T = \hat{H} + \hat{D}$) and haploid frequency ($\rho_H = \hat{H}/(\hat{H} + \hat{D})$) are obtained at the population equilibrium ($\hat{H}, \hat{D}$) (see details in

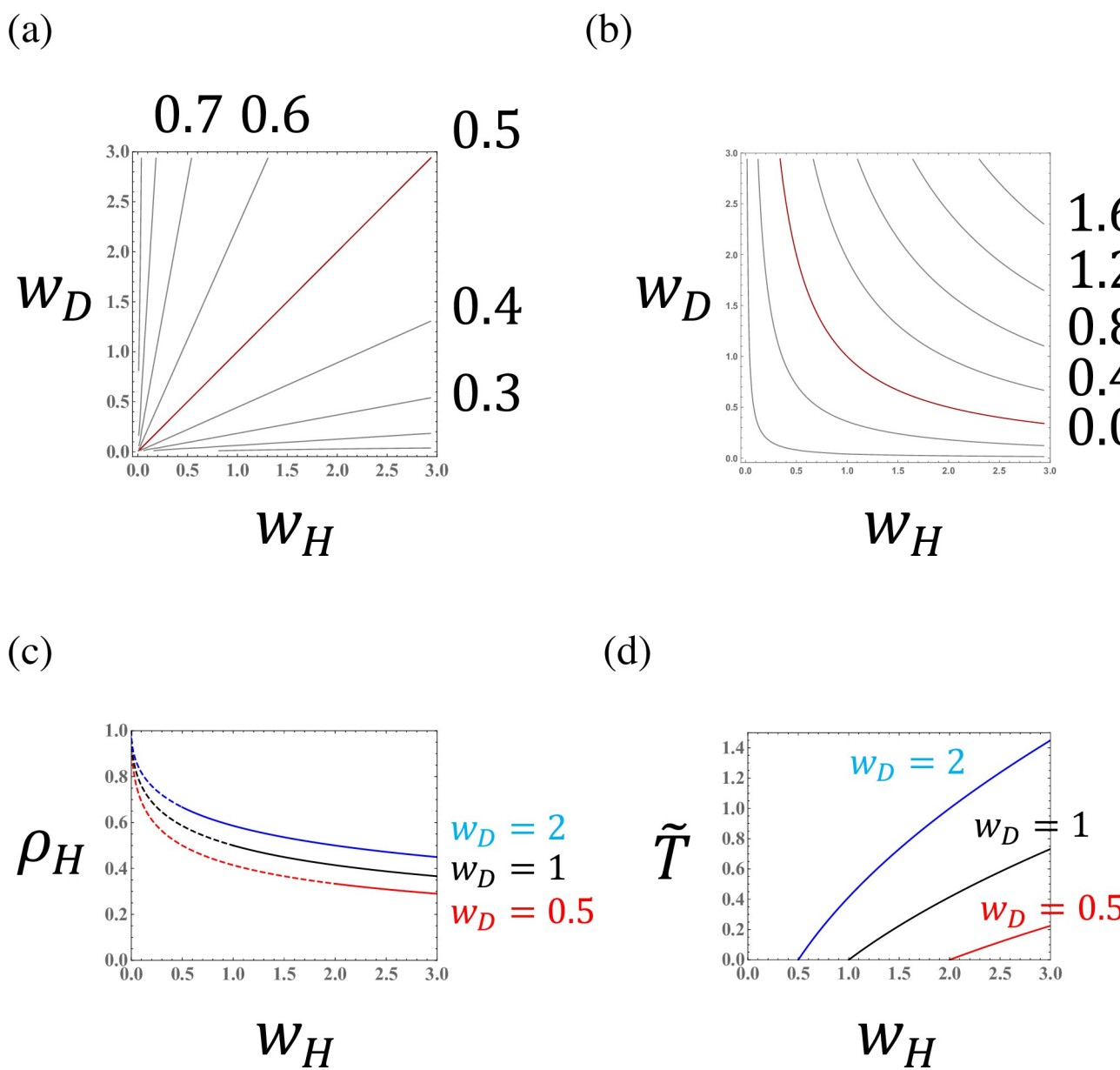

**Fig 2. The haploid frequency (($\rho_H = \hat{H}/(\hat{H} + \hat{D})$)) and the biomass measure ($\tilde{T} = R_0 - 1$) in a population of a species with obligate sexuality ($a_H = a_D = 0$).** (a) Haploid frequency curves; the red line represents the case that the haploid fraction constitutes exactly half of the population ($\rho_H = 0.5$). (b) Curves for the biomass measure. The red line represents the case that $R_0$ is exactly equal to one ($\tilde{T} = 0$). Population extinction occurs in the region where $\tilde{T}$ is negative. (c) The haploid frequency as a function of haploid fitness ($w_H$) for different values of diploid fitness ($w_D$). Dashed lines indicate values for extinct populations ($\tilde{T} < 0$). (d) The biomass measure as a function of haploid fitness ($w_H$) for different values of diploid fitness ($w_D$).

the S1 File),

$$T = \frac{1}{\delta}\left[\sqrt{\left(\frac{w_D^A - w_H^A}{2}\right)^2 + w_H^S w_D^S} + \frac{w_H^A + w_D^A}{2} - 1\right], \qquad (8A)$$

$$\rho_H = \frac{\frac{w_H^A - w_D^A}{2} - w_D^S + \sqrt{w_H^S w_D^S + \left(\frac{w_H^A - w_D^A}{2}\right)^2}}{w_H^A + w_H^S - w_D^A - w_D^S}, \tag{8B}$$

where I define the fitness components of haploids and diploids as follows:
$w_H^A = (b_H a_H \gamma_H)/m_H$, $w_D^A = (b_D a_D \gamma_D)/m_D$, $w_H^S = [b_H (1 - a_H)\sigma \gamma_D]/m_D$ and $w_D^S = [b_D (1 - a_D)\gamma_H]/m_H$.

When the fitness components of asexual reproduction are equal between haploids and diploids, $w_H^A = w_D^A$, haploid frequency is simply,

$$\rho_H = \frac{\sqrt{w_D^S}}{\sqrt{w_H^S} + \sqrt{w_D^S}}. \tag{9}$$

Interestingly, the ploidy ratio in the density-dependent model, Eq (8B), is equivalent to that in the density-independent model (3) when the mortalities of the haploid and diploid stages are equal ($m_H = m_D$).

If equal survivorships of the reproductive cells are assumed ($\gamma = \gamma_H = \gamma_D$) along with equal mortality rates ($m = m_H = m_D$), then the reproductive parameters can be simplified and the ploidy ratio at equilibrium becomes

$$\rho_H = \frac{a_H \tilde{b}_H + a_D \tilde{b}_D - 2\tilde{b}_D + \sqrt{4(1 - a_H)(1 - a_D)\sigma \tilde{b}_H \tilde{b}_D + (a_H \tilde{b}_H - a_D \tilde{b}_D)^2}}{2[a_H \tilde{b}_H + (1 - a_H)\sigma \tilde{b}_H - \tilde{b}_D]}, \tag{10}$$

where $\tilde{b}_H = (b_H \gamma)/m$ and $\tilde{b}_D = (b_D \gamma)/m$. Eq (10) is the equilibrium ploidy ratio in Bessho and Otto ([53], their Eq (A.4)). Given that they were using the Wright-Fisher model and thus ignored differences in mortality between haploids and diploids, this match makes sense.

## Parameter dependence of asexually reproducing species

For tractability, consider first the case where the fitness components of haploids and diploids are symmetrical for both sexual and asexual reproduction, $w^S = w_H^S = w_D^S$ and $w^A = w_H^A = w_D^A$. Under this condition, the basic reproductive number is the sum of the fitness components of sexual and asexual reproduction, $R_0 = w^A + w^S$. When this measure is larger than one, the trivial equilibrium is unstable, and the nontrivial equilibrium is stable (Appendix C1 in S1 Appendix).

If the assumption of symmetry is relaxed, the basic reproductive number ($R_0$) becomes somewhat more complicated (see Eq (C.6)), but Eq (7) still hold. Unfortunately, it is not possible to simplify the stability condition, so it was checked numerically. It can be concluded that this haploid-diploid system never shows bistability and oscillation. When the basic reproductive number is larger than one ($R_0 > 1$), the trivial equilibrium becomes unstable and the dynamics approaches a nontrivial equilibrium in which the haploid frequency is described by Eq (8B).

Figs 3–7 illustrate the parameter dependence of the haploid frequency ($\rho_H$) and the biomass measure ($\tilde{T} = R_0 - 1$). Both the ploidy ratio and biomass depend on the fitness components ($w_H^S$, $w_D^S$, $w_H^A$, and $w_D^A$). Because asymmetry between haploids and diploids (i.e., the cost of sex, $\sigma$) is absorbed into the transformed parameter $w_H^S$, the relationship between haploids and diploids becomes symmetrical when comparing the $w$'s.

The parameter dependence of the fitness components for sexual reproduction (Figs 3 and 4) can be understood by comparing to the case of obligate sexuality (Fig 2). As shown by Eq

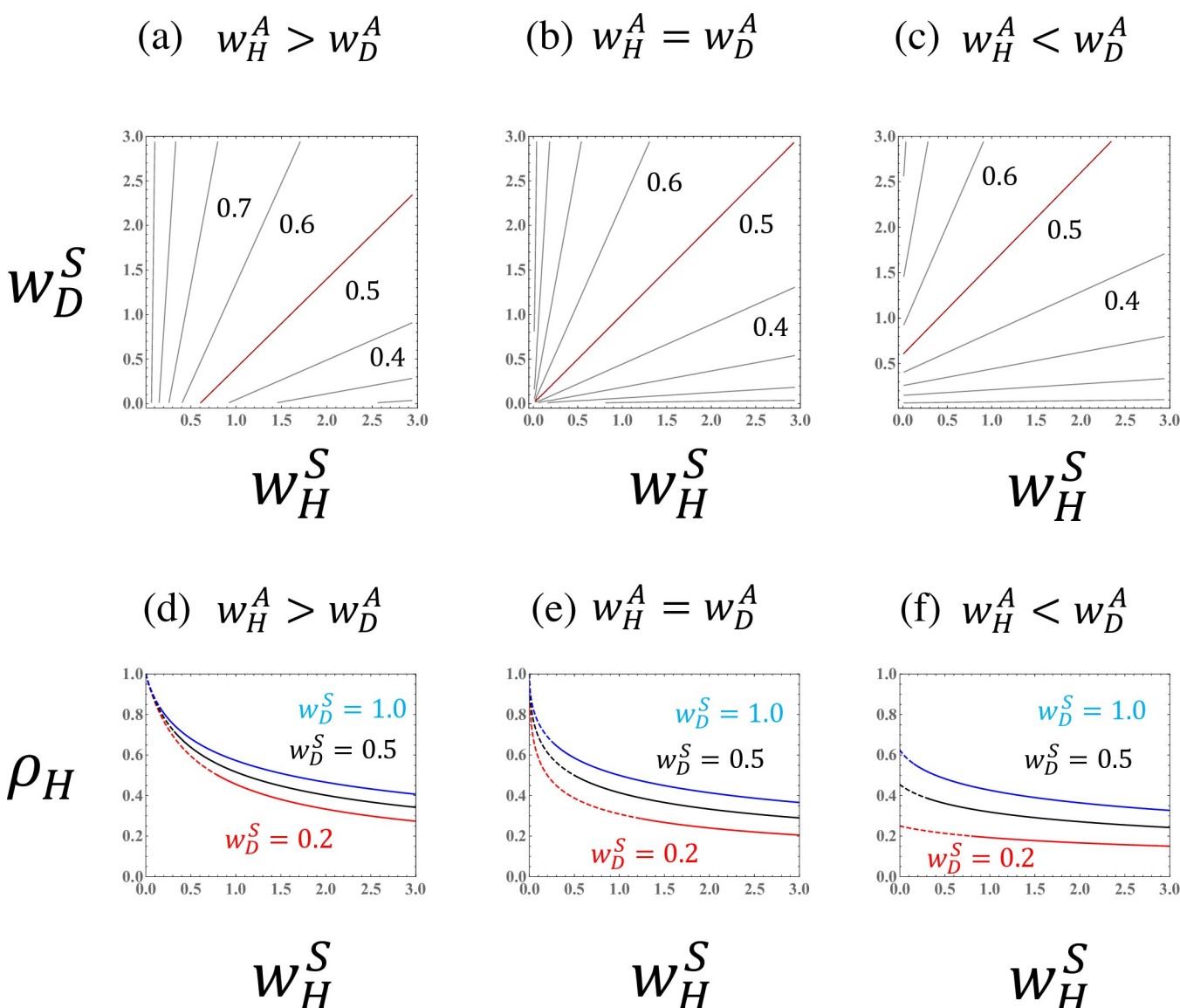

**Fig 3. Haploid frequency ($\rho_H = \hat{H}/(\hat{H} + \hat{D})$) in a population of a species with asexual reproduction when the fitness components for asexual reproduction ($w_H^A$ and $w_D^A$) are given.** (a),(b), and (c) Haploid frequency curves. The red lines represent the case where the haploid fraction constitutes exactly half of the population ($\rho_H = 0.5$). (d),(e), and (f) The haploid frequency as a function of the haploid fitness component for sexual reproduction ($w_H^S$) for different values for the diploid fitness component for sexual reproduction ($w_D^S$). Dashed lines indicate values for extinct populations ($\tilde{T} < 0$). The given parameter values are (a),(d) $w_H^A = 0.8$ and $w_D^A = 0.2$; (b),(e) $w_H^A = 0.5$ and $w_D^A = 0.5$; and (c),(f) $w_H^A = 0.2$ and $w_D^A = 0.8$.

(9), when the asexual fitness of the species is symmetrical between haploids and diploids ($w_H^A = w_D^A$), the parameter dependence is similar to that for an obligate sexual species (Figs 3B, 3E, 4B, and 4E). When the asexuality of haploids is stronger than that of diploids ($w_H^A > w_D^A$), then the region in which the haploid fraction is dominant in the population ($\rho_H > 0.5$) becomes larger. The reverse ($w_H^A < w_D^A$) is also true.

Interestingly, the haploid frequency in a population, $\rho_H$, depends on the difference in the fitness components for asexual reproduction, $w_H^A - w_D^A$, when $w_H^S$ and $w_D^S$ are fixed. Hence, the slopes of the haploid frequency curves become one (Fig 5A–5C). This result means that the ploidy ratio in a haploid-diploid population can be described by three parameters (see Eq (8B),

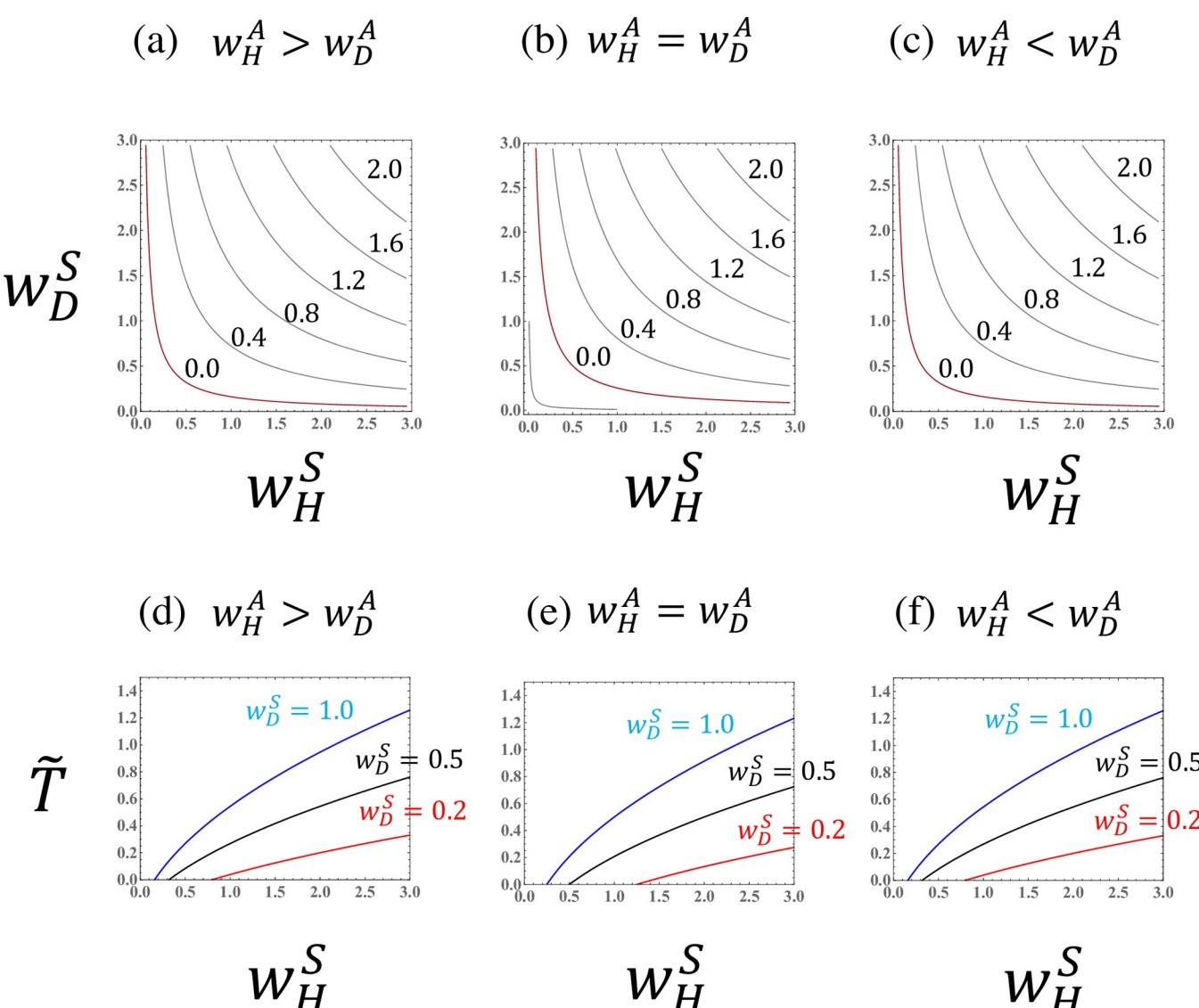

**Fig 4. The biomass measure ($\tilde{T} = R_0 - 1$) in a population of a species with asexual reproduction when the fitness components for asexual reproduction ($w_H^A$ and $w_D^A$) are given.** (a),(b), and (c) Curves for the biomass measure. The red lines represent the case that $R_0$ is exactly equal to one ($\tilde{T} = 0$). (d), (e), and (f) The biomass measure as a function of the haploid fitness component for sexual reproduction ($w_H^S$) for different values of the diploid fitness component for sexual reproduction ($w_D^S$). The given parameter values are the same as in Fig 3.

which depends on the sexual fitnesses only through $w_H^A - w_D^A$). When the haploid frequency is illustrated as a function of the difference in the fitness components for asexual reproduction ($w_H^A - w_D^A$), the curves are sigmoidal (Fig 5G).

When the fitness components for asexual reproduction of haploids and diploids are similar ($w_D^A \approx w_H^A$), then the basic reproductive number can be approximated by the sum of the geometric mean of the fitness components for sexual reproduction and the arithmetic mean of the fitness components for asexuality, $R_0 \approx \sqrt{w_H^S w_D^S} + (w_H^A + w_D^A)/2$ (see Eq (C.6)). Hence, when the fitness components for asexual reproduction are given, the biomass measure ($\tilde{T} = R_0 - 1$) is a function of the geometric mean of the fitness components for sexual reproduction (Fig 4).

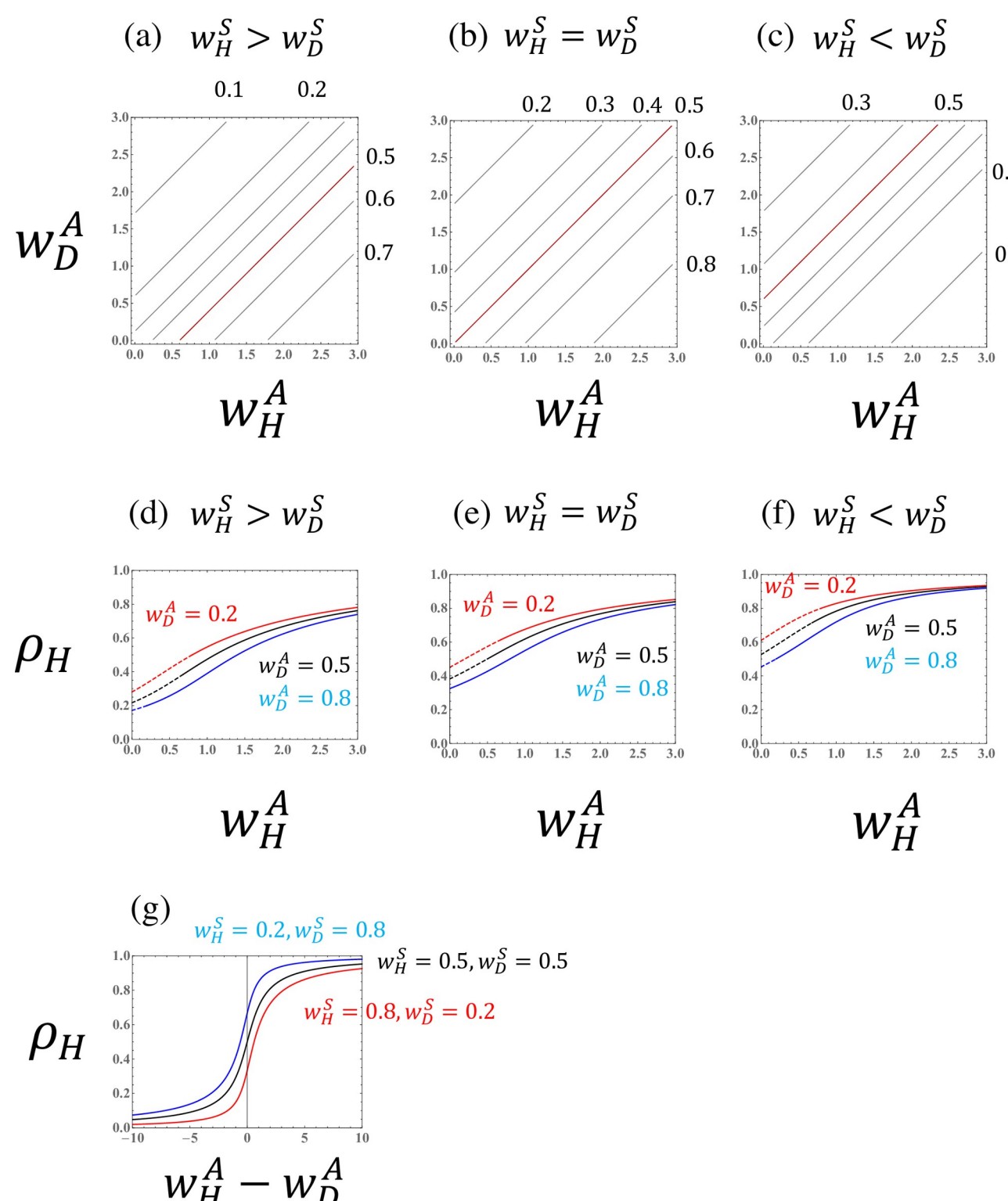

**Fig 5. The haploid frequency ($\rho_H = \hat{H}/(\hat{H} + \hat{D})$) in a population of a species with asexual reproduction when the fitness components for sexual reproduction ($w_H^S$ and $w_D^S$) are given.** (a),(b), and (c) Haploid frequency curves. The red lines represent the case that the haploid fraction constitutes exactly half of the population ($\rho_H = 0.5$). (d),(e), and (f) The haploid frequency as a function of the haploid fitness component for asexual reproduction ($w_H^A$) for different values of the diploid fitness component for asexual reproduction ($w_D^A$). Dashed lines indicate values for extinct populations ($\tilde{T} < 0$). (g) The haploid frequency as a function of the difference in the fitness components for asexual reproduction ($w_H^A - w_D^A$). The given parameter values are (a),(d) $w_H^S = 0.8$ and $w_D^S = 0.2$; (b),(e) $w_H^S = 0.5$ and $w_D^S = 0.5$; and (c),(f) $w_H^S = 0.2$ and $w_D^S = 0.8$.

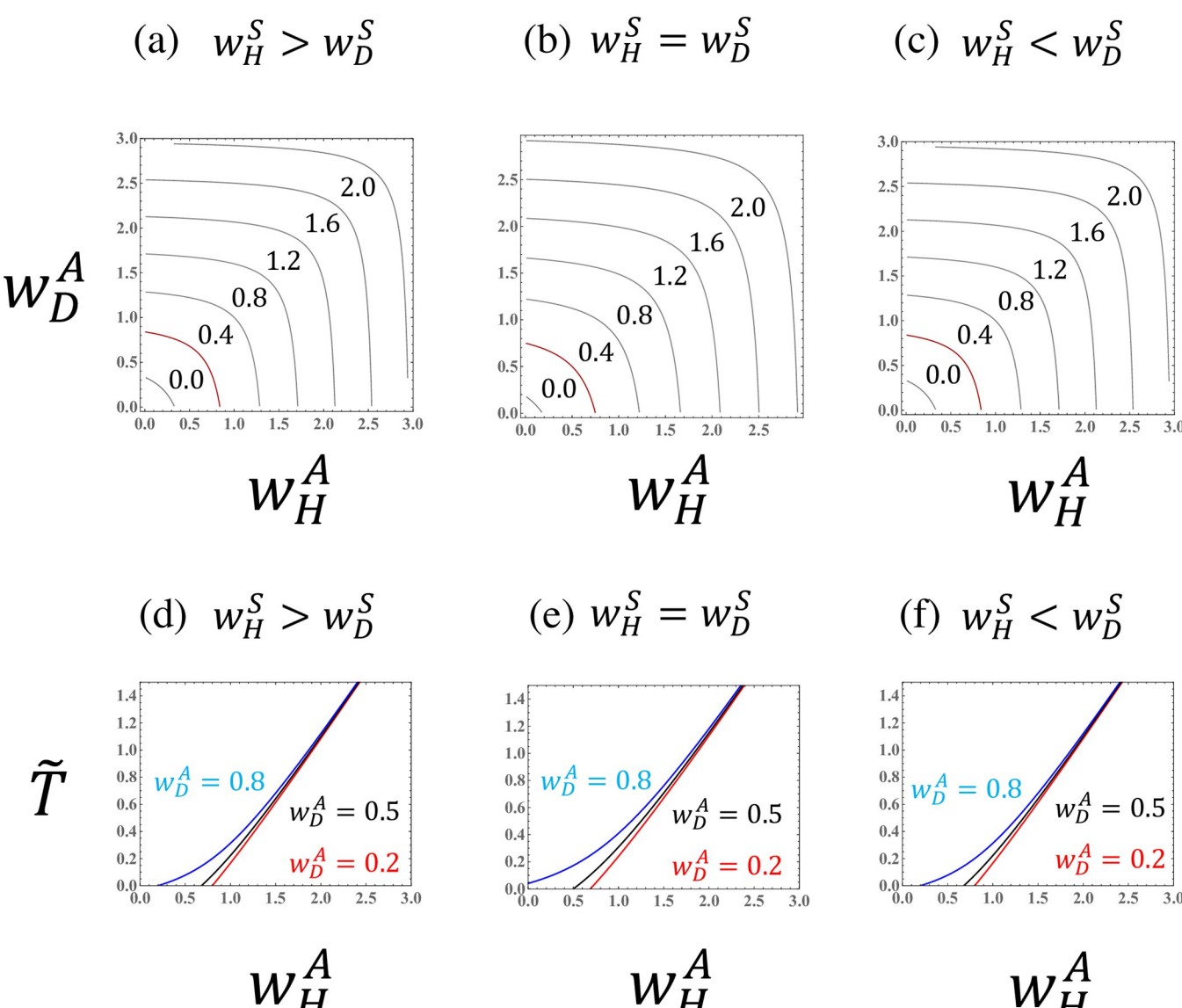

**Fig 6. The biomass measure ($\tilde{T} = R_0 - 1$) in a population of a species with asexual reproduction when the fitness components for sexual reproduction ($w_H^S$ and $w_D^S$) are given.** (a),(b), and (c) Curves for the biomass measure. The red lines represent the case that $R_0$ is exactly equal to one ($\tilde{T} = 0$). (d),(e), and (f) The biomass measure as a function of the haploid fitness component for asexual reproduction ($w_H^A$) for different values of the diploid fitness component for asexual reproduction ($w_D^A$). The given parameter values are the same as in Fig 5.

Of course, this approximation breaks down when the difference in the fitness components for asexual reproduction is large. If the fitness components for asexual reproduction are much larger than those for sexual reproduction ($w_H^A, w_D^A \gg w_H^S, w_D^S$), then the approximation, $R_0 \approx w_D^A$ (when $w_D^A > w_H^A$), holds. Thus, the biomass is dominated by the largest fitness component (Fig 7).

## Discussion

### Ploidy ratio in a haploid-diploid population

Mixtures of free-living haploid gametophytes and diploid sporophytes are widely observed in macroalgal populations. This article presents a general haploid-diploid model that explicitly

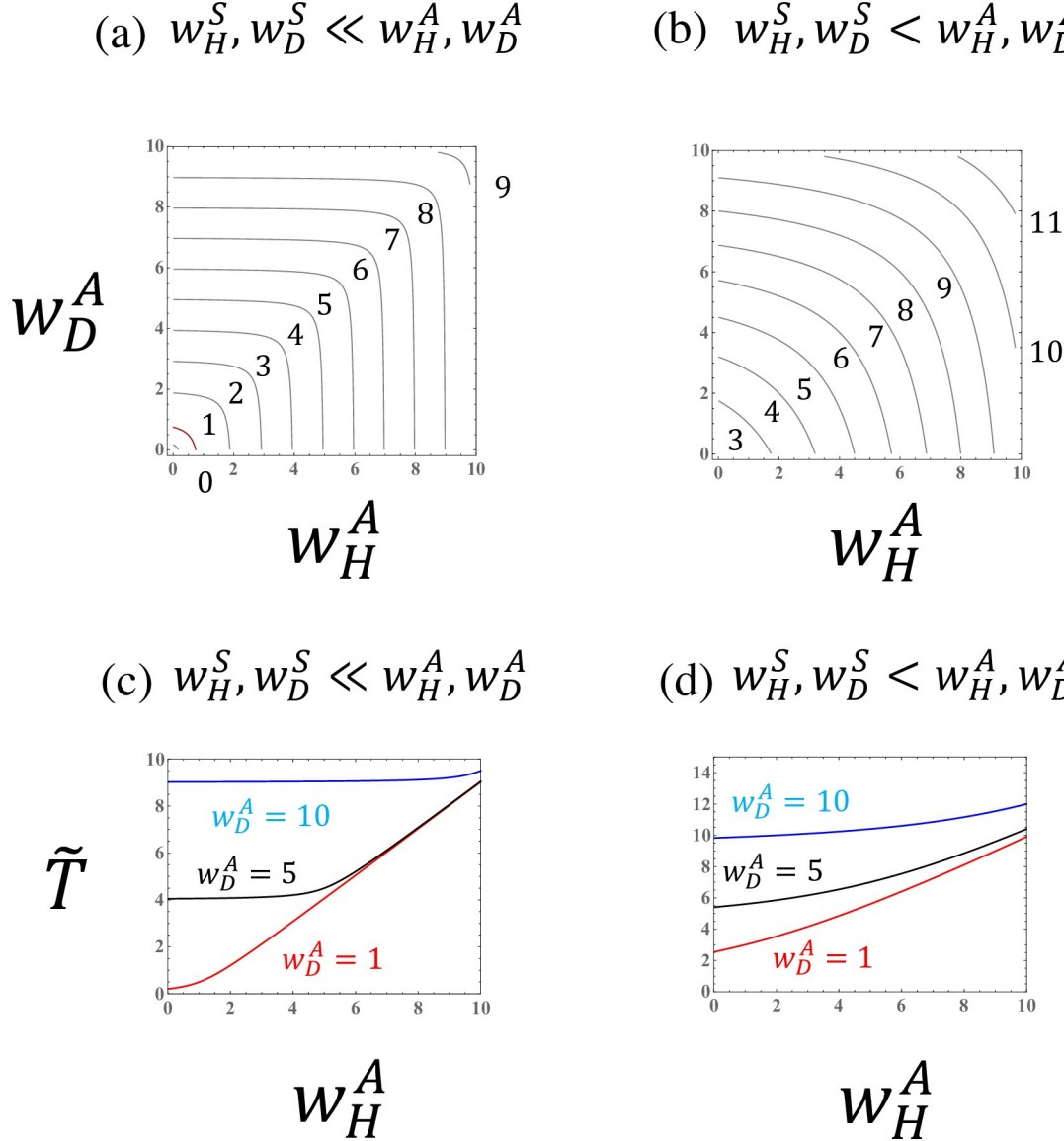

**Fig 7. The biomass measure ($\tilde{T} = R_0 - 1$) in a population of a species with asexual reproduction when the fitness components for sexual reproduction ($w_H^S$ and $w_D^S$) are given.** (a),(b) Curves of the biomass measure. (c),(d) The biomass measure as a function of the haploid fitness component for asexual reproduction ($w_H^A$) for different values of the diploid fitness component for asexual reproduction ($w_D^A$). The given parameter values are (a),(c) $w_H^S = 0.5$ and $w_D^S = 0.5$, and (b),(d) $w_H^S = 3.0$ and $w_D^S = 3.0$.

accounts for density dependence and asexual reproduction and can be used to obtain an exact solution for the ratio of haploid gametophytes to diploid sporophytes in a population at steady state. The analysis (Eq 8B) reveals that the proportions of gametophytes and sporophytes in a wild population are determined by the balance of four fitness components: sexual reproduction of haploids ($w_H^S$), sexual reproduction of diploids ($w_D^S$), asexual reproduction of haploids ($w_H^A$), and asexual reproduction of diploids ($w_D^A$). Furthermore, the difference in the asexual demographic force ($w_H^A - w_D^A$) is important. Because all life history parameters (i.e., fecundity, mortality, survivorship of reproductive cells, and the cost of sexuality) are embedded in these fitness components, the composition of a haploid-diploid population is entirely determined by

three parameters ($w_H^S$, $w_D^S$, and $w_H^A - w_D^A$). I note that the ploidy ratio that is obtained is robust to changes in the mode of competition (Appendix D in S1 Appendix).

The result is consistent with the ploidy ratios reported by previous studies for the special cases considered. For example, the classical haploid-to-diploid ratio of Thornber and Gaines [28], $\sqrt{2} : 1$, is consistent with Eq (8B) when the life history parameters of two stages are assumed to be identical ("ideal" isomorphy). The equilibrium ploidy ratios in the Wright-Fisher and Moran population genetic models [53, 60] also correspond to a special case of Eq (8B).

In summary, the model presented here generalizes the findings of previous studies [28, 30, 53, 60] and mathematically demonstrates the intuitive conclusion that the frequency of haploidy in a population is determined by the balance of fitness components, with sexual reproduction in a haploid or diploid individual leading to an increase in the other's frequency and asexual reproduction in a haploid or diploid leading to an increase in its own frequency. It should be noted, however, that the fitness component for sexual reproduction in our analysis is defined by the mortality rates of the opposite ploidy stage. For example, an increase in haploid mortality decreases the fitness component of haploid for the asexual reproduction while simultaneously decreasing the fitness component of diploid for the sexual reproduction.

## Stability and biomass of haploid-diploid populations

Despite the fact that two-dimensional systems of differential equations with density dependence may exhibit a variety of dynamics, including bistability and oscillations, previous research has lacked studies of population stability (e.g., [61]). With the help of numerical calculations, the dynamics of a haploid-diploid population has been found to exhibit a stable equilibrium state without oscillations in the parameter region in which the population exists (positive biomass).

Furthermore, the persistance of a population and the equilibrium population size depend critically on the basic reproductive number, $R_0$ (Eq (C.6)). In particular, when the fitness components for asexual reproduction of haploids and diploids are similar ($w_H^A \approx w_D^A$), then

$$R_0 \approx \sqrt{w_H^S w_D^S} + \frac{w_H^A + w_D^A}{2}. \tag{11}$$

Thus, the basic reproductive number is approximately the sum of the geometric mean of the fitness components of haploids and diploids for sexual reproduction and the arithmetic mean of the fitness components of haploids and diploids for asexual reproduction. Hence, the contribution of sexual and asexual reproduction to the existence of a population can be estimated by calculating these geometric and arithmetic means of the fitness components.

Empirical studies have shown that some macroalgal populations are maintained by asexual reproduction (e.g., [51]). Interestingly, the model presented here suggests that, when the sexual reproduction term is smaller and can be ignored, the basic reproductive number can be approximated as

$$R_0 \approx \begin{cases} w_H^A \text{ when } w_H^A > w_D^A \\ w_D^A \text{ when } w_H^A < w_D^A \end{cases}. \tag{12}$$

These results suggest that when asexual reproduction is dominant in a population, the ploidy with the larger fitness component mainly determines population persistence and total population size.

## Ecology and evolution of haploid-diploid life cycles

The life cycle diversity observed in macroalgae has been of interest in terms of the evolution of haploid versus diploid [9–11, 13]. However, a simple population genetic model proposed the problem that life cycles in which both stages are stably maintained (biphasic life cycle) do not evolve, and the importance of ecological effects began to attract attention [57, 61, 62]. As an influential study, Hughes and Otto [61] found that ecological differences between two ploidy stages can lead to the evolution and maintenance of biphasic life cycles. In contrast, the evolution of variety in haploid-diploids (i.e., heteromphy versus isomorphy, and haploid-dominant versus diploid-dominant) is still obscure. Because the evolution in a haploid-diploid should be analyzed the mutant dynamics in a resident population that reached the equilibrium (e.g., [53]), the stable ploidy ratio in this article contributes to this analysis as the first step.

The haploid-diploidy in macroalgae has raised another question for researchers: how is the ploidy ratio in the field determined? The relative importance of the difference in life history parameters (e.g., fertility, survivorship, and fertilization success) between haploids and diploids was analyzed by simple [28, 30] or complex [31, 38] mathematical models. However, our study reveals that essentially life history parameters can be summarized in only three components; sexual reproduction of haploids ($w_H^S$), sexual reproduction of diploids ($w_D^S$), the difference in the asexual demographic force ($w_H^A - w_D^A$). Of course, the measurement of individual life history parameters remains important, but in the future, when classifying and understanding the diversity of ploidy ratios observed in the field, the question of how the three components are balanced is more essential than the question of which life history traits are important. Furthermore, if the difference in asexual demographic forces is negligible, the balance between the geometric mean of sexual demographic forces ($\sqrt{w_H^S w_D^S}$) and the arithmetic mean of asexual demographic forces ($(w_H^A + w_D^A)/2$) will also be an important property in applications, in the sense that population stability and biomass can be evaluated by the sum these two means.

## Future perspectives

This article presents an analysis of the ploidy ratio and stability of macroalgal populations observed in the field. However, several issues remain to be addressed. First, it is assumed in this paper that density dependence is regulated by the population biomass. However, asymmetry between gametophytes and sporophytes may also affect the mode of competition. In other words, the question is whether the introduction of ploidy-specific competition coefficients (e.g., [61]) would change the conclusion. Similarly, since this model assumes that the parameters that determine the strength of the density dependence ($\delta$) are equivalent between haploids and diploids, the difference of $\delta$ between ploidy phases is also an interesting problem. Because $R_0$ is a measure of spread when rare determining (stability of the trivial equilibrium), it is not affected by the nature of density dependence. However, it remains unclear how the stability of a nontrivial equilibrium is affected by the competition coefficient and the strength of the density dependence; thus, it is important to extend the model to investigate these points.

Second, although it is assumed that all life-history parameters are constant, wild populations often exhibit seasonal oscillations. This issue is also related to the stability of population dynamics in the first point. It is an important task to determine whether oscillations are an intrinsic property of the population dynamics or whether they are due to parameter changes caused by seasonality and to investigate the effects of parameter changes on population dynamics.

The demography and the ploidy structure of a population are fundamental to understanding the evolution of diverse and complex life cycles. The analyses and formulas developed here

provide a basis for analyzing ecological data to discuss demography and evolution in haploid-diploid species.

## Supporting information

**S1 File. The *Mathematica* code (*Wolfram Mathematica*, ver. 10. 1. 0. 0) used for the proof is available for download as the Supplementary file.**
(ZIP)

**S1 Appendix.**
(DOCX)

## Acknowledgments

I thank Sarah Otto for helpful comments for the draft. I thank two anonymous reviewers.

## Author Contributions

**Conceptualization:** Kazuhiro Bessho.

**Data curation:** Kazuhiro Bessho.

**Formal analysis:** Kazuhiro Bessho.

**Funding acquisition:** Kazuhiro Bessho.

**Investigation:** Kazuhiro Bessho.

**Methodology:** Kazuhiro Bessho.

**Project administration:** Kazuhiro Bessho.

**Resources:** Kazuhiro Bessho.

**Software:** Kazuhiro Bessho.

**Validation:** Kazuhiro Bessho.

**Visualization:** Kazuhiro Bessho.

**Writing – original draft:** Kazuhiro Bessho.

**Writing – review & editing:** Kazuhiro Bessho.

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
