## [Decision Letter · Decision Letter 0]

7 Aug 2023

PONE-D-23-12970Stable demographic ratios of haploid gametophyte to diploid sporophyte abundance in macroalgal populationsPLOS ONE

Dear Dr. Bessho,

Thank you for submitting your manuscript to PLOS ONE. After careful consideration, we feel that it has merit but does not fully meet PLOS ONE’s publication criteria as it currently stands. Therefore, we invite you to submit a revised version of the manuscript that addresses the points raised during the review process.

We look forward to receiving your revised manuscript.

Kind regards,

Satheesh Sathianeson, Ph.D

Academic Editor

PLOS ONE

Journal Requirements:

"I thank Sarah Otto for helpful comments for the draft. This research was supported by Grants-in-Aid from the Japan Society for the Promotion of Science to KB (19K16225; 22K06407)."

"This research was supported by Grants-in-Aid from the Japan Society for the Promotion of Science to KB (19K16225; 22K06407)."

"This research was supported by Grants-in-Aid from the Japan Society for the Promotion of Science to KB (19K16225; 22K06407)."

Reviewers' comments:

Reviewer's Responses to Questions

**Comments to the Author**

1. Is the manuscript technically sound, and do the data support the conclusions?

Reviewer #1: Yes

Reviewer #2: Partly

2. Has the statistical analysis been performed appropriately and rigorously? 

Reviewer #1: Yes

Reviewer #2: Yes

3. Have the authors made all data underlying the findings in their manuscript fully available?

Reviewer #1: Yes

Reviewer #2: Yes

4. Is the manuscript presented in an intelligible fashion and written in standard English?

Reviewer #1: Yes

Reviewer #2: No

5. Review Comments to the Author

Reviewer #1: The author attempt to reconcile a long standing question about the maintenance of life cycles with more than one free-living stage using mathematical modeling. The stability of haploid and diploid stages in a population is the result of both sexual and asexual reproduction by each stage.

Specific Comments

L36: Not all algae can be considered plants. Thus, I would suggest to amend this first paragraph to focus on where haploid and diploid life cycles in such an alternation are found. In fact, they are found beyond 'plant' lineages in the Archaeplastids.

L39: Heteromorphic generations need not be marco vs. microscopic. Again, this paragraph is an over simplification of diversity.

L42: It is not a heteromorphic life cycle, but a heteromorphic alternation or heteromorphic generations.

L43: Similarly, it is not an isomorphic life cycle.

L44: strange list of citations and unclear what you mean by tend to consist of haploid and diploid?

L52-53: What does this mean?

L60: Were there differences in Chondrus in biomechanical strength?

L63: Krueger-Hadfield and Ryan (2020) JPhycol showed differences in survivorship.

L69: Please see Stoeckel et al. 2021 J Hered for asexual reproduction. This paper is not referenced in the current manuscript but deals with sexual and asexual reproduction.

L244: Switching between "i" and "we"

L261: Dependence?

L276-277: No references? This claim needs support.

Overall there is very little integration of the results or synthesis with the literature in the discussion. This is a major weakness that needs to be addressed. Moreover, there is a lack of integration across the literature with what is known in algae with regard to life cycle and reproductive system from empirical data.

Reviewer #2: I really enjoyed reading this submission. It is the kind of work that I personally like the most, both to do and to read. I am thankful to the author for giving me the pleasure of reading it.

The work is mathematically sound. Yet, for this work to be scientifically sound, it takes more than the mathematics. I have a few major criticisms that must be addressed:

The author must be clearer and more objective in his words, particularly concerning how the mathematics relate to (or model the) biology and ecology of these algae. Very important, there is a specific terminology (or nomenclature) that the author is neglecting (or misusing) and must be used (or corrected).

The work is poorly contextualized relative to these life-cycles and the most relevant theories and advances regarding their ecology and evolutionary stability.

This poor framing occurs when presenting the topic and the motive for doing this analysis, as well as when discussing the results and highlighting the key findings.

One criterion that is often used by other journals and publishers is the interest to the readers. I know from personal experience that, for the average audience in the field of algal ecology and evolution, deeply mathematical articles are not much appealing. This article is deeply mathematical and there is no way around it, nor do I wish so. Hence, for this article to be well taken by the respective scientific community, it must be very clarifying and convincing regarding the “so what?” factor. I suggest for the author to frame better this work regarding the several hypotheses advanced for the evolutionary stability of these life-cycles and unbalanced G:T ratios, and how this work advances the state-of-the art by supporting (or not) specific hypothesis.

Below, I expand on these issues and present other minor issues. Although the list is large, all of them are rather easy to solve. Once they have been adequately addressed, I can only recommend the publication of this good work.

This work must have a figure and corresponding text detailing the isomorphic biphasic life-cycle. In fact, that was the standard in most of the literature here cited. This can be either part of the introduction or a special section after the introduction and prior to the model presentation. This presentation of the life-cycle must use the correct nomenclature, including keywords as gametophytes, tetrasporophytes, carposporophytes, meiosis, mitosis, syngamy, oogonia, spermatangia, gametangia, zygote, cystocarps, tetraspores, carpospores, etc.

This description must show that the life-cycle is actually triphasic (gametophytes, carposporophytes and tetrasporophytes) and clarify how it reduces to biphasic upon the assumption of male gametophytes fertilizing all female gametangia. It should also include the alternative haploid and diploid monophasic cycles set by sporogenesis (as in Hughes and Otto, 1999) that are fundamental in the present submission. Make a clear contrast between sporogenesis and gametogenesis.

Here and elsewhere along the manuscript, the author must clarify that his “asexual reproduction” is only sporogenesis, and that he is disregarding vegetative growth (in red algae, the growth of new fronds from the same holdfast or from broken fragments).

Here and elsewhere along the manuscript, the author must be aware of the difference between fertility and fecundity. Broadly speaking, fecundity is the production of newborns. But these must survive to be recruited into the population. In this case, this comprehends spore survival while suspended in the water column, settling in suited substrate, germination and survival as germlings until juvenile thalli. Fertility is the final outcome of fecundity and spore survival.

The author should use this presentation of the life-cycle to expand on the main questions and theories about its ecology and evolution, and to frame his work relative to these questions and theories.

The article by Hughes and Otto (1999) is a very good guide to follow.

I tend to like focused, succinct text. However, this introduction is too short. It provides a vague, incomplete presentation of the topic. Sometimes it takes steps too large while overlooking issue that are important to better understand the presentation.

The author must also be clearer about haploids and diploids being phenotypically similar. Such statement (in several parts of the text) is vague. Is the author referring only to the thalli or also the spores and germlings? Regarding morphology, eco-physiology and bio-mechanics, which affect survival and growth, or also regarding fecundity and spore survival?

I dislike the idea of ecologically similar corresponding to phenotypically similar because, in my perspective, phenotype also comprehends the reproductive biology, which can never be similar among gametophytes and tetrasporophytes. Although it is perfectly understandable what the author is trying to say, I would never use such wording and I do not recall having ever red it elsewhere.

The author never presents the case that the dominance patterns (differences in abundances) follow a spatial or seasonal pattern, and that this indicates ecological and niche differentiation between haploids and diploids.

Lines 36-37: Algae are not plants.

Lines 41-44: This is a poor explanation of the specificities of heteromorphic and isomorphic biphasic life-cycles. I would merge with the previous sentences. I would first write all need be said about the heteromorphic case and only afterwards pass to the isomorphic case. In my opinion, the “going back and forth” along lines 38 to 44 is not ideal.

The sentence “In contrast, populations of species with an isomorphic life cycle tend to consist of both haploid and diploid individuals” is particularly unfortunate because both heteromorphic and isomorphic biphasic life-cycles consist of both haploid and diploid free-living individuals. The stressor here is that, in isomorphic biphasic life-cycles, haploids and diploids are apparently morphologically similar, and thus expected to also be ecologically similar and occupy the same niche. However, experimental studies have proved that gametophytes and tetrasporophytes are not exactly morphologically similar, much less eco-physiologically or bio-mechanically.

Line 46: “… field studies on [not of] the ratio of haploid-to-diploid abundances [plural] have shown both the occurrence of haploid-dominated (refs) and diploid-dominated (refs) populations”. It would also be nice to present how this ratio has been named (H:D or G:T).

Line 49-50 “haploid versus diploid dominance [not abundance] patterns”. The issue is the dominance.

Line 55: “haploids and diploids have identical phenotypes” is misleading. It was gametophytes and tetrasporophytes (i.e., the adult thalli) but not the tetraspores and carpospores (i.e., the spores, which are also either haploid or diploid). And they were compared under the assumption of identical survival and growth rates. I believe that phenotype also comprehends the reproductive biology of the respective individuals. The reproductive biology of gametophytes and sporophytes is inevitably widely different without prejudice of morphological similarity and niche overlap.

The abundance ratio (actually proposed by Destombe et al 1989 and Scrosati et al 1999, before Thornber and Gaines 2004) is a theoretical hypothesis grounded on the ecological similarity between gametophytes and tetrasporophytes and the differences between their reproductive biologies. This is not the same as the assumption that they are phenotypically similar. In fact, they cannot be phenotypically similar given that they have different reproductive structures and biologies.

This goes on throughout the text (line 57 and henceforth). My comment applies to all these instances.

Lines 59-64: All 6 lines within the same brackets is not a good way to write. Brackets should be for small intermissions. The text should be reformulated to be generally outside brackets. Krueger-Hadfield and Ryan (2020) also found differences in survival rates. Vieira et al. (2021) also found differences in growth rates. Usandizaga et al. (2023) also found differences in resistance to epiphyte infection. Thornber et al. (2006) and Verges et al. (2008) also found differences in resistance to herbivory. Many other examples exist.

Lines 65-71: “To address the shortcomings in theoretical treatments of the demography of macroalgal populations” is very vague. The author did not even debate properly the theoretical hypothesis about - and modelling of - the ecology and evolution of isomorphic biphasic life cycles, much less their shortcomings.

Lines 69-70: This is not quite true. Vieira and Santos (2010) addressed the effect of vegetative growth of new fronds from the same holdfast (it is included in what they call the looping rates). Here, and throughout the manuscript, the author only refers to sporogenesis. However, asexual reproduction also comprehends vegetative growth, fragmentation, budding, binary fission, etc.

Lines 71-72: This is Discussion and not Introduction.

Line 77: I would not brand the matrix model by Thornber and Gaines (2004) as “the classical”. Other authors have also developed matrix models of isomorphic biphasic life-cycles (ex: Engel et al, 2001; Vieira and Santos, 2010, 2012a, 2012b; Vieira and Mateus, 2014) which are not less relevant. It should be enough rephrasing to “updating the matrix model by Thornber and Gaines (2004)”.

Lines 75-93: The model presentation is mathematically correct … provided that the incorrect use of nomenclature is corrected.

“haploid and diploid reproductive cells” are actually the haploid and diploid spores, namely the tetraspores and carpospores.

“in flowing water” is while suspended in the water column.

“the survivorship of haploid and diploid reproductive cells in flowing water is ...,” is actually the survival of tetraspores and carpospores while suspended in the water column.

“the fertility of the haploid and diploid stages per individual is ...” is actually the fecundity.

Fertility is bgamma.

Adult survival is 1-m.

“Fractions aH and aD of the reproductive outputs of the haploid and diploid stages are asexual, skipping the sexual loop and developing directly into haploids and diploids.” Aside the poor English, this is sporogenesis.

Line 78: This is a poor way to start presenting the basics of the model. I propose.

“the haploid (H) and diploid (D) densities are forecasted to time t+1 from their realized densities at time t and their vital rates (equation (1)”

The Discussion does an interesting presentation of the importance of sexual and asexual reproduction, and of the new metric developed, the Reproductive number. However, it does a poor job framing these new findings in the context of the theories and questions about the ecology and evolution of this life-cycle. In fact, there are only as little as 3 citations in the Discussion, one of them being a self-citation. Several interesting questions and works are overlooked:

- the relative importance of fertility, growth and survival rates, and of differences between haploids and diploids in these vital rates (see works by Hughes and Otto, Fierst, Thornber and Gaines, Vieira, among others).

- the conditions for fixation in biphasic vs monophasic life-cycles (see works by Hughes and Otto, 1999; Hal, 2000; Bessho and Otto, 2022, among others).

- The cost of sex (see Richerd et al, 1993)

- The cost of DNA (see Lewis, 1985; Mable 2001)

6. PLOS authors have the option to publish the peer review history of their article (what does this mean?). If published, this will include your full peer review and any attached files.

---

## [Author Response · Author response to Decision Letter 0]

11 Oct 2023

Please see "Response_to_Reviewers".

---

## [Decision Letter · Decision Letter 1]

9 Nov 2023

PONE-D-23-12970R1Stable demographic ratios of haploid gametophyte to diploid sporophyte abundance in macroalgal populationsPLOS ONE

Dear Dr. Bessho,

Thank you for submitting your manuscript to PLOS ONE. After careful consideration, we feel that it has merit but does not fully meet PLOS ONE’s publication criteria as it currently stands. Therefore, we invite you to submit a revised version of the manuscript that addresses the points raised during the review process.

We look forward to receiving your revised manuscript.

Kind regards,

Satheesh Sathianeson, Ph.D

Academic Editor

PLOS ONE

Journal Requirements:

Reviewers' comments:

Reviewer's Responses to Questions

**Comments to the Author**

1. If the authors have adequately addressed your comments raised in a previous round of review and you feel that this manuscript is now acceptable for publication, you may indicate that here to bypass the “Comments to the Author” section, enter your conflict of interest statement in the “Confidential to Editor” section, and submit your "Accept" recommendation.

Reviewer #1: (No Response)

Reviewer #2: (No Response)

2. Is the manuscript technically sound, and do the data support the conclusions?

Reviewer #1: Yes

Reviewer #2: Yes

3. Has the statistical analysis been performed appropriately and rigorously? 

Reviewer #1: Yes

Reviewer #2: Yes

4. Have the authors made all data underlying the findings in their manuscript fully available?

Reviewer #1: Yes

Reviewer #2: Yes

5. Is the manuscript presented in an intelligible fashion and written in standard English?

Reviewer #1: Yes

Reviewer #2: Yes

6. Review Comments to the Author

Reviewer #1: Overall this manuscript is substantially improved and all my original comments have been addressed. However, a minor detail is to carefully go through the manuscript and choose to either use "we" or "I" as the manuscript goes back and forth between the two.

Reviewer #2: The author addressed all my comments and suggestions in a very satisfactory way. In some places, the integration of his results and analytical solutions with former literature was very well achieved (see lines 262-270 for a good example). The outcome is good and I recommend its publication. There are, nevertheless, some issues still to be addressed/corrected, from conceptualization to grammar. I detail below.

This work focuses on the effects of ploidy differences in fitness (W). The analytical solutions derived and the figures presented have W as basis. However, for W contribute sexual and asexual reproduction, but also survival (here, mortality m) (see W derivations in the text), but its effects were overlooked. Aside, growth rates were entirely neglected, and these can also be important.

Previous works showed that ploidy differences in growth and survival rates can be of utmost importance. However, here, ploidy differences in m were overlooked and the work reads as if it all depended exclusively from ploidy differences in sexual and asexual reproduction (as example, see Discusion\\ lines 362-366). This is false. The author would arrive at the same results if the ploidy differences in W were driven by ploidy differences in m. However, throughout the work the author always considers that mH and mD are equal.

Grounding my claims, a few lines below the author states “this study explicitly considers density dependence, which many previous studies have omitted for simplicity. Many previous studies (e.g., Vieira and Santos 2010; 2012) define the ploidy ratio by the right eigenvector in the matrix model, where the density dependence is ignored and the population size diverges to infinity. In reality, the population size should remain finite and the stable ploidy ratio of this population is important.”

Therefore, in the author’s own words, the factors constraining population growth and causing density-dependence are fundamental. Hence, ploidy differences in these factors are also fundamental for the ploidy ratio as well as the evolutionary stability of isomorphic biphasic life cycles. Well, the factors behind density-dependency are growth and survival, which the author presently overlooked. Even in Appendix D, if I understood well, the different modes of density-dependency are never tested considering the possibility of ploidy differences in survival/mortality.

I take this chance to clarify that the effects of density-dependency (both Allee effects (detrimental low densities) and self-thinning (detrimental high densities)), and its ploidy differences, where evaluated by Vieira et al (2022) applying an Individual Based Model to Gracilaria chilensis (aka Agarophyton chilense).

Another proof, and simultaneously a consequence, of overlooking survival are the sentences in lines 282-285: “The ploidy ratio and biomass both depend on haploid fitness () and diploid fitness (). Because haploids reproduce diploid individuals and vice versa, haploids are dominant in a population with higher diploid fitness.”

This is misleading, with two aspects contribute to it:

1) This is only true in species where their demography (and their fitness) is dominated by fertility rates, or eventually by growth rates with fecundity being proportional to frond size. However, the exact opposite happens in species for which their demography (and fitness) is dominated by survival and/or clonal growth. The current biased analysis and conclusion resulted from the author not having tested for the effects of different degrees of m and of ploidy differences in m. Fierst et al. (2005) demonstrated that the patterns of ploidy dominance are highly dependent on the amount of m. Vieira and Santos (2010), as well as proceeding works by Vieira, corroborated it and advanced by showing that for species that mainly invest in survival (i.e., low m), ploidy differences in survival (in m) or clonal growth are the dominant factor determining patterns of ploidy dominance.

2) The traditional concept of fitness is the ability of producing offspring (i.e., next generation). This is straight-forward in mono-phasic life cycles. However, in biphasic life cycles, the offspring (i.e., the next generation) is of the opposite ploidy. Hence, the fitness components comprehend the vital rates necessary to get to the opposite ploidy phase, and not to complete the full life cycle. Once this is clarified, the fitness components of the sexual loop, as presented in the text, become clear. And lines 282-285 make more sense. But if this is not clarified ahead, the reader may become confused with the W of the sexual loop and lines 282-285.

I see 2 alternatives:

1) The author clarifies that here he focuses in the reproductive aspects and in due time the author will produce further work addressing the effects of growth and survival rates. I understand that addressing them all in just one article is unfeasible. Hence, this should be the first article in a series dedicated to this study. I myself have done the same for the same reasons.

2) In the proper places throughout the text, the author clarifies that m is also a fundamental component of W. Simultaneously, the author corrects the sentences that misleadingly suggest that W is only (or primarily) determined by reproduction.

The later goes in accordance with the author argumentation in the discussion “Of course, the measurement of individual life history parameters remains important, but in the future, when classifying and understanding the diversity of ploidy ratios observed in the field, the question of how the three components are balanced is more essential than the question of which life history traits are important”. I can accept this. Just make clear throughout the text that survival/mortality is also a fundamental aspect of the fitness components (w).

other issues:

Sometimes the text is too wordy and redundant.

Lines 37-43:

“The life cycle of sexual eukaryotes is characterized by the alternation of two nuclear phases: haploid and diploid. This alternation of nuclear phases results in an alternation of generations between two ploidy stages (haploid and diploid stages) and a biphasic life cycle is observed in many plants, algae, and fungi (Raper and Flexer 1970; Willson 1981; Valero et al. 1992; Klinger 1993; Bell 1997; Mable and Otto 1998; Coelho et al. 2007). In particular, in many macroalgae, the free-living haploid gametophytes and free-living diploid sporophytes alternate (details are described in the next section).”

This first paragraph of the introduction is vague explaining the main concept but still too wordy in some sentences. The concept of biphasic life-cycle is only presented in the last sentence. However, it is debated since the first sentence. These can lead the less-experienced reader to confusion. I give an example of how it may be improved:

Sexual reproduction in eukaryotes leads to the alternation of two nuclear phases, namely the haploid (n) and diploid (2n). In many plants, algae, and fungi both ploidy stages develop into adult free-living organisms and as such, a biphasic life cycle is observed (Raper and Flexer 1970; Willson 1981; Valero et al. 1992; Klinger 1993; Bell 1997; Mable and Otto 1998; Coelho et al. 2007). Many macroalgae alternate between a free-living haploid generation (the gametophytes) and the free-living diploid generation (the sporophytes).

Line 44-45: It is redundant to say “… this alternation of generations (biphasic life cycle) …”. It was already well explained above.

Line 48-50: This sentence is confusing. Does the author intend to say that there may be species with heteromorphic biphasic life-cycles, yet with minor phenotypic differences?

Maybe the author means:

“Species with the heteromorphic life cycle show large phenotypic differences (e.g., macroscopic versus microscopic, erect versus crustose). These species also often show different seasonal appearances and habitats (Higa et al. 2007).”

Line 51: “with” instead of “exhibiting the”.

Line 55: “interesting” and not “interested”

Line 55-57: This sentence is confusing because it does not clarify that it is haploid and diploid life cycles that are being mentioned, and not haploid and diploid individuals or generations, as mentioned earlier. Also, I do not see how this sentence leads to its following sentence. The text in lines 57-61 need a better introduction to it. Overall, this paragraph needs a better introductory explanation.

Lines 69-71: “… have shown that both …”.

Also, it should be clarified that patterns of haploid or diploid dominance have been observed over different taxa, different populations of the same species, and even over the same population depending on the special location or season.

Line 94: “analyses”

Line 95-97: “… and aims to express stable ploidy ratios in a haploid-diploid population using a mathematical expression as simple as possible.”

Lines 101-104: The author should explicitly mention that hereafter “asexual reproduction” means sporogenesis. As it is, it suggests that sporogenesis is the only form of asexual reproduction, which is incorrect.

Line 105: “… analyse the right eigenvector …” is jargon too specific. Maybe “… analyse the population structure yield by a matrix model …” is more accessible to the common reader.

Line 107: Please, do not start a paragraph with “Furthermore …”

Line 133: “the brown algae Fucales, …”. Otherwise, with the “brown algae, Fucales, …” reads as if all brown algae are Fucales and have a diploid life cycle.

Lines 137-139: “Since this article focuses on the dynamics of both gametophytes and sporophytes, I assume that the alternation of generation between haploid gametophyte and diploid sporophyte always occurs.”

Besides unnecessary, this sentence is misleading and untruthful. Throughout the article, and as soon as in the Abstract, the author claims:

“Furthermore, the persistence of a haploid-diploid population and its total biomass are shown to be determined by the basic reproductive number (0), which is shown to be a function of these four demographic forces. When R0 is greater than one, the haploid- diploid population stably persists, and the ploidy ratio obtained by the analytical solution is realized”

This implies that the haploid-diploid population may not persist, with one of the phases being eliminated, namely when R0<1. This contradicts the claim above that “… I assume that the alternation of generation between haploid gametophyte and diploid sporophyte always occurs.”. No, you don’t. And you even determine the conditions for one of them to be eliminated. The conditions for such evolutionary non-stability of isomorphic biphasic life cycles have been studied before.

Lines 152-157: “I note that, however, even when there are substantial differences between gametophytes and sporophytes, they may not be extreme. For example, the species of brown algae, Ectocarpus, both gametophytes and sporophytes are small and filamentous, but there is a substantial morphological difference between them. This species is classified as “heteromorphic (near-isomorphic)” (Couceiro et al. 2015).”

I do not see the point of these text.

#1 – first sentence only says that “substantial” differs from “extreme”. Dictionaries already do that.

#2 – second sentence only says that heteromorphicity may encompass other features besides size and filamentous. That happens with other species, even of red algae, as is Asparogopsis armata and Falkenbergia rufolanosa, which are actually the haploid and diploid phases of the same life cycle.

Better English: “In the brown algae Ectocarpus, both …”

Lines 157-160: poor English. Change to “The accumulation of empirical studies showing that gametophyte and sporophyte (or tetrasporphyte of red algae) phenotypes differ (morphologically, ecophysiologically, or biomechanically) calls into question the classical classification of the life cycle into heteromorphic and isomorphic categories.”

Lines 165-16: poor English. Change to “Among red algae, however, many species from the Florideophyceae are known to exhibit more complex life cycles. Here, the female gametes fertilized on the haploid female gametophyte develop into diploid multicellular structures, the carposporophytes …”. Furthermore, this is a vague description. More detail would be preferable.

Line 168: First, the carposporophyte produces diploid carpospores by somatic cell division. Only later they are released.

Line 171-172: “three phases” and not “three alternating generations” as the carposporophyte is not free-living and thus not a generation. By calling the carposporophyte a generation, the author is contradicting himself in his former presentation of the subject (see above). “Triphasic” means 3 phases, not 3 generations.

Line 174-175: Just to make sure, I went again through the article by Kamiya and Kawai (2002), specifically dedicated to the carposporophytes. Nowhere do they call it an “independent phase”, much less “generation”. We agree that this article is the reference about carposporophytes. So, lets agree that the carposporophyte is not an “independent phase”, much less a “generation”.

Lines 203-212: The explanation of the rationale for bH/2 is vague and could be improved. The bH is the fecundity rate of haploid females (i.e., production of carposporophytes). However, not all haploids are females as some are males. Assuming a 1:1 sex ratio, ½ of the haploids are females. Hence, in the matrix model, haploid fecundity is bH/2. Then, comes the reproductive cost f, leading to fbH/2. In my opinion, this explanation is clearer than the current text.

Line 336: I believe there is o typo as W_H^S is repeated twice and there is no W_D^S.

Lines 378-383: citing those studies might be adequate.

Line 382: “… an increase in its …”

Line 388: Is this true for the work by Hughes and Otto (1999)?

Line 405: “when the sexual reproduction term is smaller and can be ignored”. This would be better placed in line 403, before the equation:

“Interestingly, the model presented here suggests that, when the sexual reproduction term is smaller and can be ignored, the basic reproductive number can be approximated as”

Line 418-419: This English is grammatically incorrect. There is number disagreement (singular and plural mixed) and it I am not sure what the author intends to say. The sentence must be revised. Maybe “… in haploid-diploid life-cycles …”?

Line 427: The Individual Based Model by Vieira et al. (2022) is the more recent and most complex of them.

7. PLOS authors have the option to publish the peer review history of their article (what does this mean?). If published, this will include your full peer review and any attached files.

Reviewer #1: No

Reviewer #2: No

---

## [Author Response · Author response to Decision Letter 1]

20 Nov 2023

Please see "Response_to_Referees" file.

---

## [Editor Report · Decision Letter 2]

22 Nov 2023

Stable demographic ratios of haploid gametophyte to diploid sporophyte abundance in macroalgal populations

PONE-D-23-12970R2

Dear Dr. Bessho,

We’re pleased to inform you that your manuscript has been judged scientifically suitable for publication and will be formally accepted for publication once it meets all outstanding technical requirements.

Kind regards,

Satheesh Sathianeson, Ph.D

Academic Editor

PLOS ONE
---

## [Editor Report · Acceptance letter]

28 Nov 2023

PONE-D-23-12970R2 

Stable demographic ratios of haploid gametophyte to diploid sporophyte abundance in macroalgal populations 

Dear Dr. Bessho:

I'm pleased to inform you that your manuscript has been deemed suitable for publication in PLOS ONE. Congratulations! Your manuscript is now with our production department. 

Kind regards, 

on behalf of

Dr. Satheesh Sathianeson 

Academic Editor

PLOS ONE